# Innovative Anti-CD38 and Anti-BCMA Targeted Therapies in Multiple Myeloma: Mechanisms of Action and Resistance

**DOI:** 10.3390/ijms24010645

**Published:** 2022-12-30

**Authors:** Danilo De Novellis, Raffaele Fontana, Valentina Giudice, Bianca Serio, Carmine Selleri

**Affiliations:** 1Department of Medicine, Surgery and Dentistry, University of Salerno, 84081 Baronissi, Italy; 2Hematology and Transplant Center, University Hospital “San Giovanni di Dio e Ruggi d’Aragona”, 84131 Salerno, Italy

**Keywords:** multiple myeloma, targeted therapy, CD38, anti-CD38 antibodies, BCMA, anti-BCMA bispecific antibodies, anti-BCMA CAR-T, mechanisms of resistance

## Abstract

CD38 and B-cell maturation antigens (BCMAs) are prevalently expressed on neoplastic plasma cells in multiple myeloma (MM), making them ideal therapeutic targets. Anti-CD38 monoclonal antibodies, such as approved daratumumab and isatuximab, are currently the milestone in MM treatment because they induce plasma cell apoptosis and kill through several mechanisms, including antibody-dependent cellular cytotoxicity or phagocytosis. BCMA is considered an excellent target in MM, and three different therapeutic strategies are either already available in clinical practice or under investigation: antibody–drug conjugates, such as belantamab-mafodotin; bispecific T cell engagers; and chimeric antigen receptor-modified T cell therapies. Despite the impressive clinical efficacy of these new strategies in the treatment of newly diagnosed or multi-refractory MM patients, several mechanisms of resistance have already been described, including antigen downregulation, the impairment of antibody-dependent cell cytotoxicity and phagocytosis, T- and natural killer cell senescence, and exhaustion. In this review, we summarize the current knowledge on the mechanisms of action and resistance of anti-CD38 and anti-BCMA agents and their clinical efficacy and safety.

## 1. Introduction

Targeted therapy is defined as an innovative type of anti-cancer treatment that includes monoclonal antibodies (MoAbs), small molecule inhibitors, antibody-drug conjugates, and immunotherapy to specifically identify and attack cancer cells while sparing normal cells and minimizing off-target side effects [1]. Multiple myeloma (MM) is a common hematological malignancy caused by the clonal proliferation of plasma cells (PCs) with a hyperproduction of monoclonal proteins (M-protein) that accumulate in the tissues and lead to organ damage [2]. Neoplastic PCs specifically express certain surface antigens compared to their normal counterparts, such as CD38 and B-cell maturation antigen (BCMA) [3], and these targets have been investigated in numerous preclinical and clinical trials. Here, we review and describe the state-of-art anti-CD38 and anti-BCMA treatments for MM, providing an update on their current use in clinical practice and mechanisms of resistance.

## 2. CD38 Functions in Health and Disease

CD38, a single chain type II transmembrane glycoprotein, is involved in lymphocyte to endothelial cells through CD31 binding, cell migration, and signal transduction [4,5], and its dysfunction impairs insulin secretion, neutrophil chemotaxis, and oxytocin release [6]. CD38 is also an ectoenzyme that regulates CD31-mediated intracellular calcium mobilization and nicotinamide adenine dinucleotide catabolism [7,8]. CD38 is expressed at low levels in myeloid, lymphoid, and other cell types, while normal and neoplastic PCs show higher surface expression of this molecule [7]. Because of its prevalent expression on PCs, CD38 represents the ideal therapeutic target for MM treatments.

## 3. Anti-CD38 MoAbs

Anti-CD38 MoAbs exert their anti-tumor action through multiple mechanisms of action, including antibody-dependent cellular cytotoxicity (ADCC), antibody-dependent cellular phagocytosis (ADCP), direct cellular apoptosis, complement-dependent cytotoxicity (CDC), and the modulation of extracellular ectoenzyme activity (Figure 1) [9]. Two anti-CD38 MoAbs, daratumumab and isatuximab, are already approved by regulatory agencies and used in clinical practice, while MOR202 and TAK-079 are under evaluation in clinical trials (Table 1).

### 3.1. Daratumumab

Daratumumab (sold under the brand name “Darzalex”), a fully humanized IgG1-κ MoAb directed against two β-strands (amino acids 233–246 and 267–280) of CD38, induces potent ADCC and CDC in MM cell lines and primary MM cells [10] and directly induces apoptosis through Fc receptor-mediated crosslinking [11] and caspase activation [12]. The clinical history of daratumumab dates back to the first-in-human phase I/II dose escalation GEN501 (NCT00574288) trial in relapsed/refractory (R/R) MM patients [13] and to the multi-center, open-label phase II SIRIUS trial (NCT01985126) evaluating intravenous daratumumab in monotherapy for R/R MM [14]. A pooled analysis of these two trials (*n* = 148) has documented an overall response rate (ORR) of 31%, including 13 very good partial responses (VGPR), four complete responses (CR), and 3 stringent CR (sCR), with a median progression-free survival (PFS) and overall survival (OS) rate of 4 and 20 months, respectively [15]. These results have led to the approval of daratumumab monotherapy, in 2015, for R/R MM previously treated with at least 3 lines of therapy, including proteasome inhibitors (PI) and/or immunomodulatory drugs (IMIDs) [16]. Intravenous daratumumab at 16 mg/kg has been associated with lenalidomide and dexamethasone (Dara-RD) (expansion cohort of GEN503, NCT01615029; and randomized open-label phase III POLLUX trial, NCT02076009), showing an ORR ranging from 88% to 93% [17,18,19,20]. Updated results of the POLLUX trial also show a benefit in PFS in high-risk cytogenetic MM (22.6 vs. 10.2 months) and standard-risk populations (not reached vs. 18.5 months) in Dara-RD cohorts [19]. Daratumumab has also been associated with bortezomib plus dexamethasone (VD) in the multicenter randomized open-label phase III CASTOR study (NCT02136134) and with pomalidomide plus dexamethasone (PD) in an open-label single-arm phase II trial (NCT01998971), showing ORRs of 83% and 66%, respectively, with minimal residual disease (MRD) negativity rates of 12% and 7%, and median PFS rates of 16.7 and 9.9 months, respectively [21,22]. The ongoing, open-label, randomized, phase III trial APOLLO (NCT03180736) is evaluating intravenous or subcutaneous daratumumab plus PD versus PD in R/R MM, showing median PFS rates of 12.4 vs. 6.9 months, respectively [23].

Intravenous daratumumab has been associated with second-generation proteasome inhibitor carfilzomib and dexamethasone (KD) (preliminary phase I trial, NCT01998971; and randomized, multicenter, open-label, phase III CANDOR trial, NCT03158688) in R/R MM with an ORR of 84% and a median PFS of 28.6 months [24,25].

Based on these good results in R/R MM, several trials have explored daratumumab efficacy and safety in newly diagnosed MM (NDMM) patients in association with bortezomib-melphalan-prednisone (VMP) (randomized controlled open-label phase III ALCYONE trial, NCT02195479) or with RD (randomized, open-label, phase 3 MAIA trial (NCT02252172), reporting ORRs of 91% and 92.9%, respectively, an MRD negativity rate of 22%, and a 36-month OS of 78% or a 30-month PFS of 70.6%, respectively [26,27,28,29,30]. Daratumumab also shows efficacy in combination with ixazomib dexamethasone, with an ORR of 87%, and nine-month PFS of 78% (prospective multi-center phase II HOVON 143 trial) [31], or with thalidomide-bortezomib-dexamethasone (VTD) in transplant-eligible NDMM patients (open-label, randomized, phase III CASSIOPEIA trial, NCT02541383) with sCR rates of 29% of cases at day 100 after autologous stem cell transplantation, and an MRD-negative status in 64% of patients [32]. These impressive and promising results, reported in the CASSIOPEIA trial, have led to the approval of daratumumab-VTD as the first line of fit transplant-eligible MM patients. 

Daratumumab is also effective in combination with lenalidomide plus bortezomib and dexamethasone (dara-VRD) in transplant-eligible MM patients (multi-center, randomized, open-label, active-controlled phase II GRIFFIN trial, NCT02874742), showing an ORR of 99% with sCR rates of up to 67%, an MRD negativity rate of 64.4%, and a 4-year PFS of 87.2% [33]. Results from the multi-center, ongoing phase III PERSEUS trial (NCT03710603), comparing daratumumab plus VRD vs. VRD in treatment-naïve MM patients, are largely expected. Moreover, daratumumab is also under evaluation in association with carfilzomib-lenalidomide-dexamethasone (KRD) as a first-line treatment in transplant-eligible MM patients [34], or with ixazomib-lenalidomide-dexamethasone as a first-line treatment for patients regardless of transplant eligibility [35].

### 3.2. Isatuximab

Isatuximab (sold under the brand name “Sarclisa”), an IgG-κ chimeric anti-CD38 MoAb, exerts a potent and distinctive Fc cross-linking-independent, lysosome-dependent pro-apoptotic activity, as well as effector functions, including CDC, ADCC, and ADCP [36,37,38]. Isatuximab has first been evaluated in xenograft models [39,40] and a phase I/II trial (NCT01084252) at a 10 mg/kg dose in R/R MM, showing better results in association with dexamethasone, with an ORR of 43.6% and a median PFS rate of 10.2 months [41]. This MoAb has also been evaluated in combination with RD for R/R MM in an open-label, dose-escalation phase Ib study (NCT01749969), showing an ORR of 56% with a good safety profile [42], or in association with PD and KD [37,43]. In the randomized, multicenter, open-label, phase III trial ICARIA-MM (NCT02990338), isatuximab plus PD has been evaluated in R/R MM patients who received ≥2 therapy lines, including lenalidomide and a PI, showing a median PFS of 17.5 months and a manageable toxicity profile [44,45]. In addition, a subgroup analysis of the ICARIA-MM trial has investigated the safety and efficacy of isatuximab-PD in patients with renal impairment, showing a longer median PFS compared to PD alone (9.5 vs. 3.7 months, respectively) [46]. Moreover, isatuximab has been evaluated in combination with KD in the randomized, multi-center, open-label IKEMA trial (NCT03275285) in R/R MM patients, displaying a CR rate of 44.1%, an MRD negativity rate of 33.5%, and a median PFS of 35.7 [47,48]. Based on the results of these trials, isatuximab in combination with PD or KD has been approved in R/R MM. 

### 3.3. MOR202 and TAK-079

MOR202 is a new human IgG-λ anti-CD38 antibody that exerts anti-MM effects through ADCC and ADCP activities [49]. MOR202 has been preliminarily evaluated in monotherapy, with dexamethasone or with dexamethasone plus one IMID in a first-in-human phase I-IIa study (NCT01421186), with a very manageable safety profile [50,51].

TAK-079, a fully human IgG1 MoAb, binds with high-affinity CD38 antigens [52]. TAK-079 is well tolerated in both intravenous and subcutaneous formulations [46] and is currently evaluated in autoimmune diseases [53] and an ongoing phase Ib trial (NCT03439280) in R/R MM patients after ≥3 lines of therapy. Updated results on TAK-079 efficacy are expected [54].

## 4. Resistance to Anti-CD38 Therapies

Several mechanisms of resistance to anti-CD38 MoAbs have been described, including CD38 downregulation, ADCC, ADCP, or CDC failure, and immune-mediated processes. High expression levels of the CD38 antigen on neoplastic PCs are essential for anti-CD38 activity [55]; however, during daratumumab treatments, CD38 levels decrease through JAK-STAT3 signaling pathway modifications [56] and return to normal levels after 3–6 months from daratumumab discontinuation [57,58]. Therefore, neoplastic clones expressing low CD38 levels could expand during treatment, and monocytes and granulocytes might favor the immune evasion of tumor cells [59]. CD38 downregulation could also be promoted by the inhibitory functions of microRNAs (miR), such as miR-26a [56,57,58,59,60]. Exposure to anti-CD38 agents also modulates the expression of genes involved in metabolism regulations and cell cycle processes [61].

Another sophisticated mechanism of resistance is the release of CD38-expressing microvesicles in the BM microenvironment after a daratumumab-promoted redistribution of CD38 on the cell surface, promoting a neoplastic PC evasion of immune surveillance [62,63,64]. These microvesicles carry high levels of immunoregulatory molecules, such as CD73, CD39, or programmed death-ligand 1 and miRNAs, and accumulate around Fc receptor-coated cells promoting immune evasion [61]. Moreover, these vesicles can be internalized and influence gene expression or can induce the production of tolerogenic adenosine [61]. Increased serum-soluble CD38 levels might also impair daratumumab efficacy, acting as a decoy receptor and altering its pharmacokinetics and pharmacodynamics [57].

Trogocytosis is an active transfer of membrane fragments containing surface antigens from presenting cells to lymphocytes within immunological synapses and is described in MM resistance through CD38 loss [63,65]. ADCC impairment is involved in resistance to daratumumab, and natural killer (NK) cell depletion causes dysfunctional ADCC [66]. This NK deficiency can be induced by several mechanisms: the direct immunosuppressive actions of neoplastic PCs; the elimination of CD38-expressing NK cells through ADCC mechanisms; and growth arrest mediated by microvesicles derived from neoplastic PCs [61,67,68,69,70]. CDC impairment can be caused by the upregulation of complement inhibitors CD55 and CD59, which reduces daratumumab-induced, complement-dependent cytotoxicity due to increased levels of CD 55 and CD 59, which is another mechanism of resistance [57].

Stromal cells in the BM niche might protect MM PCs by inducing the production of anti-apoptotic molecules [71]. Furthermore, ADCP dysfunction through CD47 amplification in MM cells is related to reduced phagocytosis and tumor escape [72,73,74,75]. In addition, low frequencies of effector T lymphocytes, with reduced expression of the costimulatory molecule CD28 and pro-inflammatory M1 macrophages, are observed in patients with R/R MM or during disease progression [63,76,77].

## 5. BCMA Functions in Health and Disease

BCMA, also known as TNFRSF17 or CD269, is a member of the tumor necrosis factor superfamily, and its natural ligands are the B-cell activating factor (BAFF) and a proliferation-inducing ligand (APRIL). BCMA is expressed in mature B lymphocytes and PCs, while it is present at low levels in other cell types [78]. Interactions between BCMA and its ligands promote MM progression, enhancing PC survival and growth through the activation of several signaling pathways, such as AKT, MAPK, and NF-kB [79]. 

Because of its selective expression on neoplastic PCs, BCMA is considered an excellent target in MM and a potential biomarker of disease monitoring and responsiveness to therapy [80,81]. Currently, three different anti-BCMA therapeutic strategies are available in MM management: antibody–drug conjugates (ADCs) (Table 2), bispecific T cell engagers (BITEs) (Table 3), and chimeric antigen receptor (CAR)-modified T cell therapies (Table 4) (Figure 2).

### 5.1. Anti-BCMA ADCs

Different from anti-CD38 MoAbs, anti-BCMA antibodies are conjugated with cytotoxic chemotherapeutic agents. A surface interaction between BCMA and the ligand first promotes drug internalization, and then the release of a chemotherapeutic agent that induces the cell death of tumoral cells [82]. Belantamab-mafodotin is the first-in-class anti-BCMA ADC; however, other drugs currently evaluated for R/R MM are AMG224, MEDI2228, and HDP-101 (Table 2).

Belantamab-mafodotin is the first-in-class humanized IgG1 anti-BCMA MoAb conjugated to the microtubule inhibitor monomethyl auristatin F. After internalization, belantamab-mafodotin induces cell death through G2/M cell cycle arrest and caspase 3-dependent apoptosis [83]. The first-in-human, open-label phase I DREAMM-1 clinical trial (NCT02064387) and the open-label, multicenter, two-arm phase II DREAMM-2 study (NCT03525678) have demonstrated impressive anti-tumor activities of belantamab-mafodotin at a dose of 3.4 mg/kg or 2.5 mg/kg every three weeks with an ORR of 60% or 34%, respectively, and median PFS of 12 or 2.8 months in heavily pretreated MM patients [84,85]. The toxicity profile is predictable and manageable, consisting of ocular complications and hematological toxicity, such as thrombocytopenia and anemia [86]. Based on these results, belantamab-mafodotin has been approved in monotherapy for R/R MM patients already exposed to four prior lines of therapy, including proteasome inhibitors, anti-CD38 MoAbs, and IMiDs. Currently, several trials are evaluating the association of belantamab-mafodotin with RD or VD (phase II DREAMM-6, NCT03544281), VRD in transplant-ineligible patients (phase III DREAMM-9, NCT04091126), inducible T cell co-stimulator agonists (aICOS) (phase I/II DREAMM-5, NCT04126200), or pembrolizumab (phase I/II DREAMM-4 NCT03848845) [87].

AMG 224 is a new anti-BCMA IgG1 antibody conjugated with mertansine (DM1), an anti-tubulin maytansinoid. The first-in-human phase I study (NCT02561962) has evaluated the pharmacokinetics and pharmacodynamics, maximum tolerated dose, and safety profile in R/R MM patients with ≥3 lines of prior therapy, including an IMiD and PI. The reported ORR is 23%, and the toxicity profile is similar to that of belantamab-mafodotin [88].

MEDI2228, an anti-BCMA fully human antibody conjugated with pyrrolobenzodiazepine (PBD) dimer, promotes DNA damage and cell death with synergistic activities with bortezomib [89,90]. The first-in-human phase I study (NCT03489525) has investigated MEDI2228 efficacy as a single agent in triple-refractory MM patients, demonstrating a maximum tolerated dose of 0.14 mg/kg every 3 weeks and an ORR of 61%. The toxicity profile is peculiar, with photophobia (54%), thrombocytopenia (32%), rash (30%), increased gamma-glutamyl transferase (24%), dry eye (20%), and pleural effusion (20%) [91]. 

HDP-101, an anti-BCMA conjugated with α-amantin, interferes with RNA polymerase II subunit A, thus inhibiting cellular transcription. HDP-101 showed pre-clinical activity in myeloma cell lines, with a preferential effect in cells with the deletion of chromosome 17. An early phase I in-human trial of HDP-101 is currently ongoing (NCT04879043) [92].

### 5.2. Anti-BCMA/CD3 Bispecific MoAbs

Bispecific MoAbs serve as an innovative therapeutic strategy already approved in hematological malignancies, including acute lymphoblastic leukemia [93], and are currently under investigation in MM. Bispecific MoAbs first recruit CD3+ immune effector T cells, and then BCMA+ neoplastic PCs, leading to TCR-independent T cell activation and neoplastic PC death through granzyme and perforin secretion [94,95]. To date, no bispecific antibodies are approved by regulatory agencies for MM treatment, even though several pre-clinical and clinical trials are evaluating their efficacy and safety (Table 3).

Teclistamab (JNJ-64007957), a fully humanized IgG4 anti-BCMA/CD3 bispecific MoAb, has shown activity in pre-clinical and phase I-II studies (NCT03145181 and NCT04557098) in R/R MM, with an ORR of 63%, an MRD negativity rate of 26.7%, and a median PFS of 11.3 months [96,97]. Drug-related toxicity includes infections, neutropenia, anemia, thrombocytopenia, grade I/II cytokine release syndrome, and neurological events [98,99]. Several ongoing clinical studies are evaluating the efficacy and safety of teclistamab in association with other anti-MM drugs in R/R MM (NCT04108195, NCT05243797, NCT05083169, and NCT04722146).

PF-06863135 (PF-3135 or elranatamab), a humanized IgG2a anti-BCMA/CD3 bispecific MoAb, has a dose-dependent action [100] and is currently evaluated in the ongoing phase I MagnetisMM-1 trial (NCT03269136) in patients with pluri-relapsed MM. Elranatamab is subcutaneously administered at different dosage schedules every 7 or 14 days in R/R MM patients, showing a preliminary ORR of 64% [101]. Elranatamab is also under investigation in monotherapy and in association with other anti-MM regimens (NCT05090566, NCT04649359, NCT05317416, NCT05020236, NCT04798586, and NCT05228470).

AMG 420, an investigational BiTE, binds BCMA on MM cells, resulting in T cell-mediated cytotoxicity through the Fas pathway [102]. R/R MM patients treated with AMG 420 in the first-in-human, dose-escalation, phase I trial (NCT02514239) showed an ORR of 30%, with a median duration of response of 9 months. The most common side effects are grade I/II cytokine release syndrome and infections [103]. Despite these promising results, the discomfort of continuous intravenous administration led to the suspension of its development.

REGN5458, an anti-BCMA/CD3 MoAb, induces the T cell-mediated killing of MM cells in vitro and the inhibition of tumor growth in mouse models [104]. The phase 1, dose-escalating trial LINKER-MM1 (NCT03761108) is currently recruiting R/R MM patients who received at least 3 prior therapy regimens. REGN5458 has demonstrated good and durable clinical activity in seven MM patients (ORR, 53.3%) without safety concerns [105].

CC-93269 (EM801), an asymmetric, double-arm, humanized IgG T cell-recruiter MoAb, bivalently binds to BCMA and monovalently to CD3 antigens with dose-dependent anti-tumor activity [106]. Indeed, no R/R MM patients treated with <3 mg responded to therapy in the first human trial (NCT03486067), while subjects treated with 3–6 mg and >6 mg had ORRs of 36% and 89%, respectively [107].

TNB-383B, an anti-BCMA/CD3 bispecific MoAb, has a double anti-BCMA arm with a silenced human IgG4 Fc region and a 10-day half-life in animal models [108]. TNB-383B induces PC death, a dose-dependent T cell activation, reduced cytokine production, and tumor growth arrest [109,110]. The open-label, multicenter, phase 1 trial NCT03933735 is currently ongoing to evaluate its safety, efficacy, and pharmacokinetics in R/R MM patients treated with at least 3 prior regimens. However, no interim results have been reported yet [111].

AMG 701 is similar to AMG 240 with a longer half-life, and promotes potent T cell activation in vitro, with synergistic effects with IMIDS [112]. A clinical phase I trial (NCT03287908) is currently ongoing to evaluate AMG 701 efficacy and safety.

## 6. Anti-BCMA CAR-T Cells

CAR-T cell treatment is a breakthrough innovative strategy in the management of hematologic malignancies, consisting of engineered autologous T cells for the recognition of tumor cells [113,114]. Adverse effects are caused by immune system hyperactivation and include cytokine release syndrome (CRS), immune effector cell-associated neurotoxicity syndrome (ICANS), cytopenia, and infections [115]. Neurocognitive and hypokinetic movement disorders (parkinsonisms) after anti-BCMA CAR-T cell infusion are novel emergence cell therapy-related adverse events and are likely caused by an autologous immune attack against BCMA-expressing neurons and astrocytes in the caudate nucleus [116]. Currently, only idecabtagene vicleucel and ciltacabtagene autoleucel are approved by regulatory agencies for clinical use (Table 4).

Idecabtagene vicleucel (ide-cell; bb2121), an anti-BCMA T cell product, is composed of autologous T cells transfected with a lentiviral vector for the expression of a murine anti-BCMA fragment, a 4-1BB co-stimulatory domain, and a CD3 activation motif. Idecabtagene vicleucel effectively kills in vitro neoplastic PCs regardless of BCMA expression levels [86]. The multi-center phase I trial (NCT02658929) evaluating different bb2121 doses in R/R MM patients who failed at least 3 therapy lines has shown an ORR of 85% with a negative MRD status and a median PFS of 11.8 months. The toxicity profile is similar to that of other CAR-T cell therapies, including neutropenia, anemia, CRS, and neurological toxicity [117]. These results have been confirmed in the pivotal phase II KarMMa clinical trial (NCT03361748) conducted in R/R MM, showing an ORR of 73% or 81.5% and a median PFS of 8.8 or 11.3 months, based on the number of infused cells. Negative MRD status has been achieved in 26% of all treated patients. CAR-T cell expansion occurs at a median of 11 days, and a more intense expansion is associated with deeper responses. CAR-T cells can still be detected 12 months after infusion [118]. KARMMA-2 (NCT03601078) is an ongoing, multicohort, phase II study designed to explore the role of bb2121 in R/R MM (cohort 1), including patients with fewer prior therapy lines characterized by worse prognosis, such as an early progression (within 18 months) from the previous treatment [autologous stem cell transplantation (cohort 2a) or not (cohort 2b)] or an unsatisfactory response after an autologous stem cell transplantation (cohort 2c). The phase I KARMMA-4 (NCT04196491) trial evaluates bb2121 efficacy in high-risk R-ISS III NDMM following standard induction. The phase III KARMMA-3 study (NCT03651128) will compare patients with R/R MM randomized to receive bb2121 or the standard of care.

Ciltacabtagene autoleucel (cilta-cel-JNJ-68284528 or JNJ-4528, previously named LCAR-B38M) is a peculiar second-generation CAR-T product because of the presence of two different heavy-chain variable domains recognizing separate epitopes of BCMA antigens [119]. In the single-arm, open-label, phase I/II LEGEND-2 trial (NCT03090659), enrolled R/R MM patients showed an ORR of 88%, a negative MRD status in 63% of cases, and a median PFS of 20 months for all patients and 28 months for MRD-negative subjects [119,120]. In the phase Ib/II CARTITUDE-1 trial (NCT03548207), a single JNJ-4528 dose of 0.75 × 10^6^ per kg was infused 5–7 days after lymphodepletion, showing an ORR of 97%, a 12-month PFS and OS of 77% and 89%, respectively. CRS and neurological toxicity have occurred in 95% (grade III–V: 4%) and 21% (grade III–IV: 9%) of cases [121,122]. The first update of the phase II CARTITUDE-2 study has demonstrated an ORR of 88.9% in R/R MM patients with an MRD negativity rate of 100% [123]. The phase III CARTITUDE-4 (NCT04181827) study is currently ongoing and aims to compare JNJ-4528 to conventional treatments (PVd: pomalidomide + bortezomib + dexamethasone; or DPd: daratumumab + pomalidomide + dexamethasone) in R/R MM. These ongoing (CARTITUDE-5, NCT04923893 and CARTITUDE-6, NCT05257083) trials will provide insights into the use of JNJ-4528 after VRD induction for the treatment of naïve MM patients not planned for autologous stem cell transplantation and will compare the efficacy and safety of this strategy for VRD induction followed by RD (CARTITUDE-5) or to daratumumab-VRD induction followed by autologous stem cell transplantation (CARTITUDE-6).

CT053, a second-generation CAR-T product, consists of a fully human anti-BCMA single-chain fragment variant, a 4-1BB co-stimulatory domain, and a CD3-zeta signaling domain. Single-arm, open-label, 3-site phase I trials (NCT03716856, NCT03302403, and NCT03380039) are currently ongoing to assess CT053 safety and efficacy in R/R MM, showing a toxicity profile similar to that of CAR-T cell therapies and an ORR of 87.5% [90]. In addition, results from the phase I/II LUMMICAR (NCT03975907) and phase Ib/II LUMMICAR-2 (NCT03915184) trials reported an impressive ORR of 100% and a 12-month PFS of 85.7% in R/R MM (median prior therapy lines, 6) [124,125,126].

Orvacabtagene autoleucel (orva-cel-JCARH125), a fully human CAR-T cell product with a 4-1BB costimulatory domain, is currently being investigated in the multi-center phase I/II EVOLVE trial (NCT03430011) in R/R MM patients treated with at least 3 prior regimens. JCARH125 can induce an ORR in 92% of treated subjects with classic and manageable safety profiles [127]. 

CART-BCMA, another CAR-T cell product expressing BCMA-specific CAR with tandem TCR and 4-1BB costimulatory domains produced by a lentiviral system, is currently under evaluation in the open-label, single-center, phase I pilot study (NCT02546167), showing lower ORRs compared to other CAR-T cell trials with a higher incidence of severe side effects [128].

The novel second-generation P-BCMA-101 CAR-T product is produced by the piggyBac™ (PB) DNA modification system rather than viral vectors, requiring only plasmid DNA and mRNA with lower production costs and the creation of a purified CAR-T population [129]. Moreover, P-BCMA-101 contains a Centyrin™, a fully human protein with high specificity and binding affinities with smaller, more stable, and potentially less immunogenic activities compared to traditional single-chain variable fragments [130]. P-BCMA-101 has been evaluated in a 3 + 3, dose-escalation, phase I trial (NCT03288493) in R/R MM, showing a very low incidence of adverse effects, including CRS and neurotoxicity, and high efficacy, with an ORR of 83% [130]. Based on these favorable preliminary results, the pivotal phase II PRIME trial (NCT03288493) is ongoing, and no hospital admission is required due to the very low rate of severe and non-severe adverse effects [131]. 

CT103 is a second-generation, fully human, BCMA-specific CAR-T cell product explored in an open-label, single-arm, phase I trial (ChiCTR1800018137). R/R MM patients treated with this cell therapy showed an ORR of 100% and a 12-month PFS of 58.3%, which is higher in those subjects without extramedullary disease (79.1%) [132]. Updated results from the single-arm, open-label, multicenter, ongoing, phase I/II FUMANBA-1 study (NCT05066646) showed an ORR of 94.9% with MRD negativity in R/R MM patients treated with 5 median prior therapy lines. CT103A also has a manageable toxicity profile [133]. A multi-center, single-arm, phase I FUNAMBA-2 trial (NCT05181501) is investigating the role of CT103A after induction therapy in high-risk NDMM. 

MCARH171, a second-generation, human-derived CAR-T cell product containing a BCMA scFv and a 4-1BB co-stimulatory domain, including a truncated epidermal growth factor receptor safety system, has been evaluated in the first-in-human, dose-escalation, phase I study (NCT03070327) in R/R MM, achieving an ORR of 64% and a median duration of response of 106 days [134].

KITE-585 is an autologous, fully human, anti-BCMA CAR-T specifically targeting BCMA-expressing MM cells [135]. The first-in-human, open-label, multicenter, phase I trial NCT03318861 has investigated efficacy and safety in R/R MM subjects, showing no grade III/V CRS or ICANS. Unfortunately, 64.3% of patients experienced disease progression, 21% experienced disease stability, and only 10% experienced a partial response [136].

## 7. Bispecific CARs

PCs can lose CD19 antigen expression during differentiation [99]; however, neoplastic PCs can retain their expression, thus explaining the potential use of anti-CD19 CAR-T cell therapy in MM [137,138]. Therefore, bispecific CAR-T cells targeting both BCMA and CD19 have been designed to improve tumoral killing. In a preliminary clinical evaluation in R/R MM, all treated patients achieved a response to treatment, with no severe side effects observed [139].

BM38 is another bispecific CAR-T product targeting BCMA and CD38 antigens with 4-1BB signaling and CD3ζ domains. In the Chinese dose-escalating, phase I trial (ChiCTR1800018143) in R/R MM, patients treated with BM38 showed an ORR of 87%, with an MRD negativity status in 87.5% of cases and a median PFS of 17.2 months [140].

## 8. CAR-NK Cells

NK cells, a subset of innate immune system cells involved in anti-viral and anti-tumor responses, have high cytotoxic activities against different types of cells, and low potential for graft-versus-host disease after allogeneic bone marrow transplantation [141,142]. Therefore, NK cells are increasingly used for immunotherapy because of their low immunogenicity, low risk of CRS, and low ICANS incidence [143,144,145]. Currently, only a few phase I/II (NCT03940833) or early phase I (NCT05008536) trials are investigating the role of anti-BCMA CAR-NK cells in patients with R/R MM [146,147,148].

## 9. Resistance to Anti-BCMA Therapies

The anti-BCMA therapeutic approach is relatively recent and, consequently, mechanisms of resistance are still poorly understood.

Biallelic or monoallelic *BCMA* loss on chromosome 16p has been observed in patients treated with anti-BCMA CAR-T cell product, ide-cel [149,150]. Point *BCMA* mutations have been described as another mechanism of resistance to BCMA therapies [151]. BCMA density on the PC surface is also related to a lack of response and might result from the selective immune pressure exerted by CAR-T cell therapies [152,153]. Furthermore, BCMA can be cleaved from the cell surface by a γ-secretase, that releases a soluble BCMA, acting as a decoy receptor for anti-BCMA treatments [154]. Another mechanism of resistance is the induction of T cell senescence or exhaustion [155,156]. The clinical significance of these mechanisms of resistance to anti-BCMA agents raises concerns regarding whether or not sequential anti-BCMA therapies can be used, even though the few reported cases confirm the efficacy of sequential approaches [157,158,159]. However, larger prospective trials are needed to better define the role of this strategy.

Mechanisms of resistance to CAR-T cells are under investigation, and published data from the single-cell transcriptomic analyses of a single PC leukemia patient treated with anti-BCMA CAR-T cells have shown gene expression modification after therapy involving genes related to proliferation, cytotoxicity, and intracellular signaling pathways [160]. Moreover, the use of bispecific CAR-T cells (e.g., anti-BCMA/CS1) could overcome monospecific CAR-T cell therapy resistance by greatly increasing tumor cell recognition and killing and by reducing the risk of antigen escape [161].

## 10. Conclusions

Despite the introduction of several immunomodulatory agents and targeted therapies in clinical practice, MM, a malignant PC disorder, remains an incurable disease with a high rate of relapse, even after hematopoietic stem cell transplantation. CD38 and BCMA are excellent therapeutic targets in MM because of their prevalent expression on neoplastic PCs, and multiple types of targeted therapies have been developed, such as MoAbs, ADCs, bispecific antibodies, and CAR-T cells. Based on their impressive results in phase II/III trials in multi-refractory patients—with ORRs higher than 90%—these treatments can dramatically change MM outcomes in the near future, and accelerated approvals have frequently been granted [162,163]. Moreover, other promising therapies still under clinical trial investigation, including bispecific antibodies, are largely expected to be promising treatments because of their clinical efficacy and safety in preliminary phase I/II trials. However, as per the third Newton’s law of action and reaction, once a biological target is under “attack” by a selective drug, this target reacts, thus developing a mechanism of resistance. Therefore, despite the dramatic and promising clinical efficacy and safety of anti-CD38 and anti-BCMA agents, we still do not exactly know how neoplastic cells can escape from drug-induced tumor killing. These mechanisms pose a future challenge in terms of how and when these novel targeted therapies should be used during the long management of an incurable disease, such as MM, where the patient experiences multiple disease relapses and high rates of refractoriness to therapy. Therefore, the best drug combination and timing should be identified to have the highest synergistic effects on tumor cell killing and to reduce the relapse rate. To date, two anti-CD38 MoAbs, daratumumab and isatuximab, are used in early therapeutic lines in both transplant-eligible and ineligible MM patients in real-life settings [27,29,32,46,47]. Belantamab-mafodotin is currently the only approved anti-BCMA ADC for pluri-refractory MM patients [85], and ciltacabtagene autoleucel and idecabtagene vicleucel are the currently approved CAR-T cell therapies [164,165].

In conclusion, MM is still virtually incurable; however, innovative anti-CD38 and anti-BCMA drugs, especially CAR-T cell therapies, might completely revolutionize MM outcomes, which might become a chronic, curable disease in the near future.

## Figures and Tables

**Figure 1 ijms-24-00645-f001:**
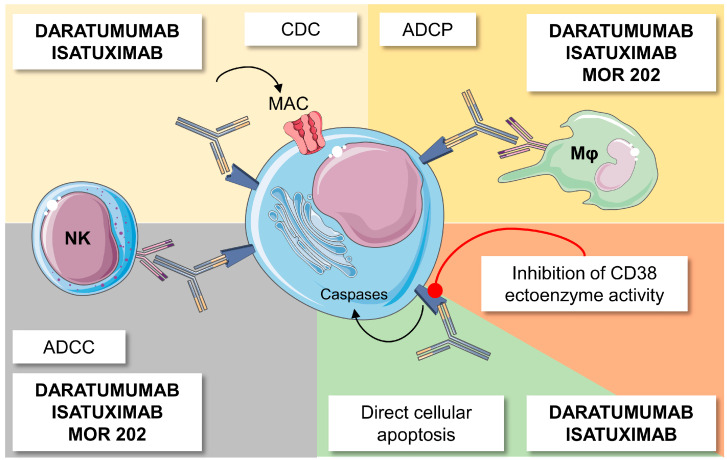
Mechanisms of action of anti-CD38 agents. Anti-CD38 monoclonal antibodies induce complement-dependent cytotoxicity (CDC) with the formation of a complement membrane attack complex leading to cell lysis; antibody-dependent cellular phagocytosis (ADCP) mediated by macrophages (Mφ); antibody-dependent cellular cytotoxicity (ADCC) mainly mediated by natural killer (NK) cells and cytotoxic T cells; direct cellular apoptosis; and modulation of extracellular ectoenzyme activity. Made using smart.servier.com.

**Figure 2 ijms-24-00645-f002:**
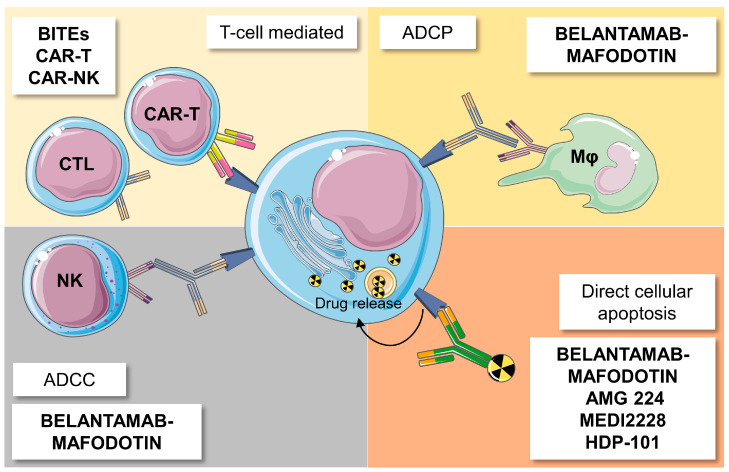
Mechanisms of action of anti-BCMA agents. Anti-BCMA agents can induce antibody-dependent cellular phagocytosis (ADCP) mediated by macrophages (Mφ); antibody-dependent cellular cytotoxicity (ADCC), mainly mediated by natural killer (NK) cells; direct cellular apoptosis through the intracellular release of cytotoxic agents; and tumor cell killing by cytotoxic T cells (CTL) and engineered chimeric antigen receptor (CAR)-T cells. Made using smart.servier.com.

**Table 1 ijms-24-00645-t001:** Anti-CD38 MoAb clinical trials.

MoAb	Trial	Phase	Disease	N. pt	Treatment	Compared Treatment
Daratumumab	NCT00574288 (GEN 501)	I/II	R/R MM	72	Dara	
	NCT01985126 (SIRIUS)	II	R/R MM ≥3 lines	124	Dara	
	NCT01615029 (GEN 503)	I/II	R/R MM	32	Dara-RD	
	NCT02076009 (POLLUX)	III	R/R MM	569	Dara-RD	RD
	NCT02136134 (CASTOR)	III	R/R MM	498	Dara-VD	VD
	NCT01998971	I	R/R MM ≥2 lines	102	Dara-PD	
	NCT03180736 (APOLLO)	III	R/R MM	304	Dara-PD	PD
	NCT01998971 (EQUULEUS)	I	R/R MM	85	Dara-KD	
	NCT03158688 (CANDOR)	III	R/R MM	466	Dara-KD	KD
	NCT02195479 (ALCYONE)	III	TN MM unfit patients	706	Dara-VMP	VMP
	NCT02252172 (MAYA)	III	TN MM unfit patients	737	Dara-RD	RD
	HOVON 143	II	MM frail patients	46	Dara-IxD	
	NCT02541383 (CASSIOPEIA)	III	TN MM fit patients	1085	Dara-VTD	VTD
	NCT02874742 (GRIFFIN)	II	TN MM fit patients	292	Dara-VRD	
	NCT03710603 (PERSEUS)	III	TN MM fit patients	690	Dara-VRD	VRD
Isatuximab	NCT01084252	I/II	R/R MM ≥3 lines	164	Isa	
	NCT01749969	Ib	R/R MM	57	Isa-RD	
	NCT02283775	Ib	R/R MM ≥2 lines	45	Isa-PD	
	NCT02990338 (ICARIA-MM)	III	R/R MM ≥2 lines	307	Isa-PD	PD
	NCT03275285 (IKEMA)	III	R/R MM	302	Isa-KD	KD
MOR202	NCT01421186	I/IIa	R/R MM	91	MOR202	MOR202-RD/PD
TAK-079	NCT03439280	Ib	R/R MM ≥3 lines	34	TAK-079	

Abbreviations. MoAb, monoclonal antibody; R/R MM, relapsed/refractory multiple myeloma; dara, daratumumab; R, lenalidomide; D, dexamethasone; V, bortezomib; P, pomalidomide; K, carfilzomib; TN, treatment naïve; M, melphalan; Ix; Isa, isatuximab.

**Table 2 ijms-24-00645-t002:** Anti-BCMA ADC clinical trials.

Drug	Trial	Phase	Disease	N. pt	Treatment	Compared Treatment
Belantamab-mafodotin	NCT02064387 (DREAMM-1)	I	R/R MM ≥3 lines	73	BelMaf	
	NCT03525678 (DREAMM-2)	II	R/R MM ≥3 lines	97	BelMaf	
	NCT03544281 (DREAMM-6)	I/II	R/R MM	152	BelMaf-VD	BelMaf-RD
	NCT04091126 (DREAMM-9)	III	TN MM unfit patients	144	BelMaf-VRD	VRD
	NCT03848845 (DREAMM-4)	I/II	R/R MM ≥3 lines	41	BelMaf + pembrolizumab	
AMG 224	NCT02561962	I	R/R MM ≥3 lines	41	AMG-224	
MEDI2228	NCT03489525	I	R/R MM ≥3 lines	82	MEDI2228	
HDP 101	NCT04879043	I	R/R MM ≥3 lines	78	HDP-101	

Abbreviations: BCMA, B-cell maturation antigen; ADC, antibody–drug conjugates; R/R MM, relapsed/refractory multiple myeloma; BelMaf, belantamab-mafodotin; R, lenalidomide; D, dexamethasone; V, bortezomib; TN, treatment naïve.

**Table 3 ijms-24-00645-t003:** Anti-BCMA/CD3 bispecific MoAb clinical trials.

Drug	Trial	Phase	Disease	N. pt	Treatment	Compared Treatment
Teclistamab	NCT03145181 (MajesTEC-1)	I	R/R MM ≥3 lines	157	Tecl s.c.	Tecl i.v.
	NCT04557098 (MajesTEC-1)	II	R/R MM ≥3 lines	192	Tecl s.c.	
	NCT04108195	Ib	R/R MM ≥3 lines	295	Dara + Tecl s.c.	Dara + P + Tecl s.c.
	NCT05243797 (MajesTEC-4)	III	R/R MM	1000	Tecl s.c. + R	R
	NCT05083169 (MajesTEC-3)	III	R/R MM	630	Dara + Tecl s.c.	Dara-PD/RD
	NCT04722146 (MajesTEC-2)	I	R/R MM	146	Tecl s.c. + others	
Elranatamab	NCT03269136 (MagnetisMM-1)	I	R/R MM ≥3 lines	103	Elrat i.v.	Elrat s.c./Elrat + P/Elrat + R
	NCT05090566 (MagnetisMM-4)	II	R/R MM ≥3 lines	105	Elrat + Niro	Elrat-RD
	NCT04649359 (MagnetisMM-3)	II	R/R MM ≥3 lines	187	Elrat	
	NCT05317416 (MagnetisMM-7)	III	MRD^+^ after auto-HSCT	366	Elrat	R
	NCT05020236 (MagnetisMM-5)	III	R/R MM	589	Elrat	Elrat+Dara/Dara-PD
	NCT04798586 (MagnetisMM-2)	I	R/R MM ≥3 lines		Elrat	
	NCT05228470 (MagnetisMM-8)	II	R/R MM ≥3 lines	36	Elrat	
AMG 420	NCT02514239	I	R/R MM ≥2 lines	43	AMG 420	
REGN5458	NCT03761108 (LINKER-MM1)	I/II	R/R MM ≥3 lines	291	REGN5458	
CC93269	NCT03486067	I	R/R MM	220	CC93269	
TNB-383B	NCT03933735	I/II	R/R MM ≥3 lines	214	TNB-383B	
AMG 701	NCT03287908	I	R/R MM ≥3 lines	408	AMG 701	AMG 701 + P/ AMG 701 + PD

Abbreviations: BCMA, B-cell maturation antigen; MoAb, monoclonal antibody; R/R MM, relapsed/refractory multiple myeloma; Tecl, teclistamab; s.c., subcutaneous; i.v., intravenous; dara, daratumumab; P, pomalidomide; R, lenalidomide; D, dexamethasone; Elrat, elranatamab; MRD, minimal residual disease; HSCT, hematopoietic stem cell transplantation.

**Table 4 ijms-24-00645-t004:** Anti-BCMA CAR-T cell clinical trials.

Drug	Trial	Phase	Disease	N. pt	Treatment	Compared Treatment
Idecabtagene vicleucel	NCT02658929	I	R/R MM ≥3 lines	33	Ide-cel	
	NCT03361748 (KarMMa)	II	R/R MM ≥3 lines	140	Ide-cel	
	NCT03601078 (KARMMA-2)	II	High-risk R/R MM	235	Ide-cel	
	NCT04196491 (KARMMA-4)	I	High-Risk TN MM	13	Induction + Ide-cel	
	NCT03651128 (KARMMA-3)	III	R/R MM 2–4 lines	381	Ide-cel	S.o.C.
Ciltacabtagene autoleucel	NCT03090659 (LEGEND-2)	I/II	R/R MM ≥3 lines	57	Cilta-cel	
	NCT03548207 (CARTITUDE-1)	Ib/II	R/R MM ≥3 lines	113	Cilta-cel	
	NCT03758417 (CARTIFAN-1)	II	R/R MM ≥3 lines	130	Cilta-cel	
	NCT04133636 (CARTITUDE-2)	II	TN or R/R MM	18	Cilta-cel	Cilta-cel + others
	NCT04181827 (CARTITUDE-4)	III	R/R MM 1–3 lines	419	Cilta-cel	Dara-VD/PD
	NCT04923893 (CARTITUDE-5)	III	TN MM Unfit patients	650	VRD + Cilta-cel	VRD + RD
	NCT05257083 (CARTITUDE-6)	III	TN MM	750	Dara-VRD + Cilta-cel	Dara-VRD + Auto-HSCT
CT053	NCT03716856, NCT03302403 and NCT03380039	I	R/R MM ≥2 lines	24	CT053	
	NCT03975907 (LUMMICAR)	I/II	R/R MM ≥3 lines	114	CT053	
	NCT03915184 (LUMMICAR-2)	I/II	R/R MM ≥3 lines	105	CT053	
JCARH125	NCT03430011 (EVOLVE)	I/II	R/R MM ≥3 lines	169	JCARH125	
CART-BCMA	NCT02546167	I	R/R MM ≥3 lines	25	CART-BCMA	
P-BCMA-101	NCT03288493 (PRIME)	I/II	R/R MM	135	P-BCMA-101	
CT103	ChiCTR1800018137	I	R/R MM	18	CT103	
	NCT05066646 (FUMANBA-1)	I/II	R/R MM ≥3 lines	132	CT103	
	NCT05181501 (FUMANBA-2)	I	High-risk TN MM	20	Induction + CT103	
MCARH171	NCT03070327	I	R/R MM ≥2 lines	20	MCARH171	
KITE-585	NCT03318861	I	R/R MM ≥3 lines	17	KITE-585	

Abbreviations: BCMA, B-cell maturation antigen; CAR, chimeric antigen receptor; R/R MM, relapsed/refractory multiple myeloma; TN, treatment naïve; S.o.C., standard of care; s.c., subcutaneous; V, bortezomib; dara, daratumumab; P, pomalidomide; R, lenalidomide; D, dexamethasone; HSCT, hematopoietic stem cell transplantation.

## Data Availability

Not applicable.

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
