# Peer review of "Innovative Anti-CD38 and Anti-BCMA Targeted Therapies in Multiple Myeloma: Mechanisms of Action and Resistance"

_ijms, 2022, doi:10.3390/ijms24010645_

Round 1
Reviewer 1 Report
The authors summarize the results of clinical trials of anti-CD38 and anti-BCMA targeted therapies, and describe an overview of the mechanisms of action and resistance of these therapies.
Overall, this review does not include recent findings about the mechanism of action and resistance.
The authors should cite recent papers and provide more comprehensive insights into this topic. For example, the authors just said, “Natural Killer cell depletion is related to dysfunctional ADCC.” But recent papers showed the mechanisms of NK cell depletion and provided the insight of how to overcome this depletion (Blood. 2020 136(21):2416-2427, Clin Cancer Res. 2021 27(10):2947-2958 Blood Adv. 2021 5(15):3021-3031) Other resistant mechanisms including CD38 downregulation (Blood. 2020 36(20):2334-2345, Cancer Res. 2020 80(10):2031-2044), immune cell exhaustion, immune evasion and others (Br J Haematol. 2021 193(3):581-591) should be more deeply discussed.
The same can be said of BCMA. For example, recent studies examined CAR-T dynamics in patients and provided mechanistic insights (Mol Ther. 2021 29(2):645-657), and also showed how to prevent antigen escape in CAR-T therapy (Nat Commun. 2020 11(1):2283) In clinical aspect, BCMA therapy have the potential risk of neurotoxicity other than ICANS because BCMA is expressed in some neurons and astrocytes (Nat Med. 2021 27(12):2099-2103).
Finally, although the authors said, “Anti-BCMA CAR-T treatments are ---- they are not currently approved by regulatory agencies”, anti-BCMA CAR-Ts (Ciltacabtagene autoleucel and Idecabtagene vicleucel) have been already approved and used for myeloma patients worldwide. The authors should review and update entire manuscript.
Author Response
The authors summarize the results of clinical trials of anti-CD38 and anti-BCMA targeted therapies, and describe an overview of the mechanisms of action and resistance of these therapies.
Comment 1. Overall, this review does not include recent findings about the mechanism of action and resistance.
The authors should cite recent papers and provide more comprehensive insights into this topic. For example, the authors just said, “Natural Killer cell depletion is related to dysfunctional ADCC.” But recent papers showed the mechanisms of NK cell depletion and provided the insight of how to overcome this depletion (Blood. 2020 136(21):2416-2427, Clin Cancer Res. 2021 27(10):2947-2958 Blood Adv. 2021 5(15):3021-3031) Other resistant mechanisms including CD38 downregulation (Blood. 2020 36(20):2334-2345, Cancer Res. 2020 80(10):2031-2044), immune cell exhaustion, immune evasion and others (Br J Haematol. 2021 193(3):581-591) should be more deeply discussed.
Response to Comment 1. We apologize for poor information on mechanisms of resistance to anti-CD38 agents. Suggested references were also added.
On pages 5-6, lines 152-189, the section has been reorganized as follows.
“4. Resistance to anti-CD38 therapies
Several mechanisms of resistance to anti-CD38 MoAbs have been described, including CD38 downregulation, ADCC, ADCP, or CDC failure, and immune-mediated processes. High expression levels of CD38 antigen on neoplastic PCs are essential for anti-CD38 activity [55]; however, during daratumumab treatments, CD38 levels de-crease through JAK-STAT3 signaling pathway modifications, and return to normal levels after 3-6 months from daratumumab discontinuation [56-57]. Therefore, neo-plastic clones expressing low CD38 levels could expand during treatment, and monocytes and granulocytes might favor immune evasion of tumor cells [58]. CD38 down-regulation could be also promoted by inhibitory functions of microRNAs (miR), such as miR-26a [62]-[63]. Exposure to anti-CD38 agents also modulates expression pf genes involved in metabolism regulations and cell cycle processes [64].
Another sophisticated mechanism of resistance is the release of CD38-expressing microvesicles in the BM microenvironment after a daratumumab-promoted redistribution of CD38 on cell surface, promoting neoplastic PCs evasion of immune surveil-lance [59-61]. These microvesicles carry high levels of immunoregulatory molecules, such as CD73, CD39, or programmed death-ligand 1, and miRNAs, and accumulate around Fc receptor-coated cells promoting immune evasion [64]. Moreover, these vesicles can be internalized and influence gene expression, or can induce the production of tolerogenic adenosine [64]. Increased serum soluble CD38 levels might also impair daratumumab efficacy, acting as decoy receptor and altering its pharmacokinetics and pharmacodynamics [56].
Trogocytosis is an active transfer of membrane fragments containing surface antigens from presenting cells to lymphocytes within immunological synapses, and is described in MM resistance through CD38 loss [60],[65]. ADCC impairment is in-volved in resistance to daratumumab, and Natural Killer (NK) cell depletion causes dysfunctional ADCC [66]. This NK deficiency can be induce by several mechanisms: direct immunosuppressive actions of neoplastic PCs; elimination CD38-expressing NK cells through ADCC mechanisms; and growth arrest mediated by microvesicles de-rived from neoplastic PCs [64][67][68]–[70]. CDC impairment can be caused by upregulation of complement inhibitors CD55 and CD59, that reduces daratumumab-induced complement-dependent cytotoxicity, due to increased levels of CD 55 and CD 59, is another mechanism of resistance [56].
Stromal cells in the BM niche might protect MM PCs by inducing the production of anti-apoptotic molecules [71]. Furthermore, ADCP dysfunction through CD47 amplification in MM cells is related to reduced phagocytosis and tumor escape [72]–[75] In addition, low frequencies of effector T lymphocytes with reduced expression of the costimulatory molecule CD28 and of pro-inflammatory M1 macrophages are observed in patients with R/R MM or at disease progression [60],[76][77].”
Comment 2. The same can be said of BCMA. For example, recent studies examined CAR-T dynamics in patients and provided mechanistic insights (Mol Ther. 2021 29(2):645-657), and also showed how to prevent antigen escape in CAR-T therapy (Nat Commun. 2020 11(1):2283) In clinical aspect, BCMA therapy have the potential risk of neurotoxicity other than ICANS because BCMA is expressed in some neurons and astrocytes (Nat Med. 2021 27(12):2099-2103).
Thank you for your comments. We added this sentence in the paragraph N. 7 of anti-BCMA CAR-T
A very interesting and novel type of neurological toxicity was the occurrence of a neurocognitive and hypokinetic movement disorder (Parkinsonism) after anti-BCMA CAR-T cell infusion. This would appear to have been related to destruction of the basal ganglia by CAR-T cells due to BCMA expression on neurons and astrocytes in the caudate nucleus [115].
Response to Comment 2. We apologize for poor information on mechanisms of resistance to anti-BCMA agents. Suggested references were also added.
On page 13, lines 453-459, the following text was added “Mechanisms of resistance to CAR-T cells are under investigation, and published data from single-cell transcriptomic analysis of a single PC leukemia patient treated with anti-BCMA CAR-T cells has shown gene expression modification after therapy involving genes related to proliferation, cytotoxicity, and intracellular signaling path-ways [149]. Moreover, the use of bispecific CAR-T cells (e.g., anti-BCMA/CS1) could overcome monospecific CAR-T cell therapy resistance by greatly increasing tumor cell recognition and killing and by reducing the risk of antigen escape [156].”
Comment 3. Finally, although the authors said, “Anti-BCMA CAR-T treatments are ---- they are not currently approved by regulatory agencies”, anti-BCMA CAR-Ts (Ciltacabtagene autoleucel and Idecabtagene vicleucel) have been already approved and used for myeloma patients worldwide. The authors should review and update entire manuscript.
Response to Comment 3. We apologize for this incorrect information, and we have changed the text accordingly.
On page 10, lines 315-320, the following text was added “Neurocognitive and hypokinetic movement disorders (parkinsonisms) after anti-BCMA CAR-T cell infusion are novel emergence cell therapy-related adverse events, and are likely caused by an autologous immune attack against BCMA-expressing neurons and astrocytes in the caudate nucleus [116]. Currently, only idecabtagene vicleucel and ciltacabtagene autoleucel are approved by regulatory agencies for clinical use (Table 4).”
Reviewer 2 Report
This a a very comprehensive and up-to-date review of antibody-based therapeutic approaches for multiple myeloma, focused on anti-CD38 and anti-BCMA targeted therapies. The presented data is accurate and covers all the aspects of the field, from "naked" antibodies, to ADCs, CAR-T and CAR-NK cells. My only very minor criticism should be resolved by minor editing of the Engligh language and minor spell-check. For eamaple, in the Abstract (second line), instead of "making them as ideal therapeutic targets" it should be written: "making them ideal therapeutic targets". In the Introduction, line 29-30, instead of "is a common he-29 matology malignancy" write: "is a common hematologycal malignancy".
I recommend that in section 3.1, after "Daratumumab" write "sold under the brand name "Darzalex" and in section 3.2, after "Isatuximab" write: "Sold under the brand name "Sarclisa".
I don't have any other comments.
Author Response
This a very comprehensive and up-to-date review of antibody-based therapeutic approaches for multiple myeloma, focused on anti-CD38 and anti-BCMA targeted therapies. The presented data is accurate and covers all the aspects of the field, from "naked" antibodies, to ADCs, CAR-T and CAR-NK cells. My only very minor criticism should be resolved by minor editing of the Engligh language and minor spell-check. For example, in the Abstract (second line), instead of "making them as ideal therapeutic targets" it should be written: "making them ideal therapeutic targets". In the Introduction, line 29-30, instead of "is a common he-29 matology malignancy" write: "is a common hematologycal malignancy".
I recommend that in section 3.1, after "Daratumumab" write "sold under the brand name "Darzalex" and in section 3.2, after "Isatuximab" write: "Sold under the brand name "Sarclisa".
I don't have any other comments.
Response to General Comments. We really thank the Reviewer for this positive feedback and for suggestions. We have checked our manuscript for English editing, and we have implemented sections 3.1 and 3.2 as suggested.
Round 2
Reviewer 1 Report
The authors answered my comments and improved the manuscript.